

# Constraining the timing and processes of pediment formation and dissection: implications for long-term evolution in the Western Cape, South Africa

Janet C. Richardson[1], Veerle Vanacker[2], David M. Hodgson[3], Marcus Christl[4], Andreas Lang[5]

[1]Geography and Geology: Department of History, Geography and Social Sciences, Edge Hill University, Ormskirk, L39 4QP, UK

[2]Earth and Life Institute, Centre for Earth and Climate Research, Université catholique de Louvain, Louvain-la-Neuve, 1348, Belgium

[3]School of Earth and Environment, University of Leeds, Leeds, LS2 9JT, UK

[4]Ion Beam Physics, ETH Zürich, Zürich, Otto-Stern-Weg 5, CH 8093, Switzerland

[5]Department of Geography and Geology, Universität Salzburg, Salzburg, A-5020, Austria

*Correspondence to*: Janet C. Richardson (Janet.Richardson@edgehill.ac.uk)

**Abstract.** Pediment surfaces are a widespread feature of the southern African landscape and have long been regarded as ancient landforms. Cosmogenic nuclide data from four pediment surfaces in the Gouritz catchment, Western Cape, South Africa are reported, including boulder surface samples and a depth profile through a colluvial pediment deposit. The results indicate low surface lowering rates (0.315 to 0.954 m My$^{-1}$) and minimum exposure ages of 0.678 – 4.462 My (assuming denudation rates of 0.3 m My$^{-1}$). Duricrusts have developed in the pediments and are preserved in some locations, which represent an internal geomorphic threshold limiting denudation and indicate at least 1 My of geomorphic stability following pediment formation. The pediments and the neighbouring Cape Fold Belt are deeply dissected by small order streams that form up to 280 m deep river valleys in the resistant fold belt bedrock geology, indicating a secondary incision phase of the pediments by these smaller order streams. Using the broader stratigraphic and geomorphic framework, the minimum age of pediment formation is considered to be Miocene. Several pediment surfaces grade above the present trunk valleys of the Gouritz River, which suggests that the trunk rivers are long-lived features that acted as local base levels during pediment formation and later incised pediments to present levels. The geomorphic processes controlling the formation and evolution of the pediments varied over time; with pediments formed by hillslope diffusive processes as shown by the lack of fluvial indicators in the colluvial deposits and later development by fluvial processes with small tributaries dissecting the pediments. Integrating various strands of evidence indicates that the pediments are long-lived features. Caution should be taken when interpreting cosmogenic nuclide ages from pediment surfaces in ancient landscapes, as isotopic steady state conditions can be reached.

## 1 Introduction



Recent advancements in geochronology allow erosion rates and exposure ages of landforms to be established, and to place
more precise constraints on landscape evolution. Establishing erosion rates and landform ages is essential for linking the
evolution of drainage systems to downstream aggradation processes (e.g. Gallagher and Brown, 1999; Chappell et al., 2006;
Tinker et al., 2008a; Wittmann et al., 2009; Sømme et al., 2011; Romans et al., 2016), constraining surface uplift and tectonic
processes (e.g., Brook et al., 1995; Burbank et al., 1996; Granger et al., 1997; Jackson et al., 2002; Wittmann et al., 2007;
Bellin et al., 2014; Vanacker et al., 2015), and palaeo-climate reconstructions (e.g., Margerison et al., 2005; Dunai et al., 2005;
Owen et al., 2005; Willenbring and Blackenburg, 2010). Reconstructing ancient landforms and landscape development is
challenging due to fragmented preservation and increasing signal overprinting forming a landscape palimpsest (e.g. Chorley
et al., 1984; Bloom, 2002; Bishop, 2007; Jerolmack and Paola, 2010; Richardson et al., 2016). However, ancient landscapes
and landforms cover a large portion of the globe (e.g., (1) Australia – e.g., Ollier, 1991, Ollier and Pain, 2000, Twidale, 2007
a,b; (2) southern South Africa – e.g., Du Toit, 1954, King 1956a, (3) South America – e.g. King, 1956b, Carignano et al., 1999,
Demoulin et al., 2005, Panario et al., 2014, Peulvast and Bétard, 2015; (4) Asia – e.g., Gorelov et al., 1970, Gunnell et al.,
2007, Vanacker et al., 2007; and (5) Europe – e.g., Lidmar-Bergström, 1988, Bessin et al., 2015) and offer important insights
into long-term Earth surface dynamics and landscape evolution (indicating variation in erosion and deposition). Further,
pediments and planation surfaces can offer insights into mantle dynamics as they are characterised by undulations with middle
(several tens of kms) to very long wavelengths (several thousands of kms) characteristic of lithospheric and mantle
deformations (e.g., Braun et al., 2014; Guillocheau et al. 2018).

The formation of pediments is contentious and four categories of landscape evolution models (Fig.1) exist that address the
evolution of pediments and surrounding mountain belts (Dohrenward and Parsons, 2009) (1) range front retreat where
channelised fluvial processes are dominant (e.g., Gilbert, 1877; Paige, 1912; Howard 1942); (2) range front retreat where
diffuse hillslope and peidmont processes are dominant (e.g., Lawson, 1915; Rich; 1935; Kesel, 1977; Bourne and Twidale,
1998; Dauteuil et al., 2015); (3) range front retreat as a result of fluvial and diffusive erosion processes (e.g., Bryan, 1923;
Sharp, 1940); and (4) lowering of the range due to channelised flow, catchment development and fluvial incision (e.g., Lustig,
1969; Parsons and Abrahams, 1984). Model type 1 acknowledges the occurrence of diffusive processes and model type 2
acknowledges the occurrence of channelised erosion processes, but each model argues these are subsidiary formation processes
(Gilbert, 1877; Rich, 1935; Howard, 1942). Model type 3 integrates fluvial and diffusive erosion processes, and their relative
importance depends on the geomorphic setting (Bryan, 1923; Sharp, 1940) with dominance of diffusive processes in regions
with erosion-resistant bedrock lithologies, ephemeral streams and a low range. Model type 4 is associated with drainage basin
development in the range, and does not require parallel retreat of the mountain front to form the pediment surfaces (Lustig,
1969; Parsons and Abrahams, 1984).



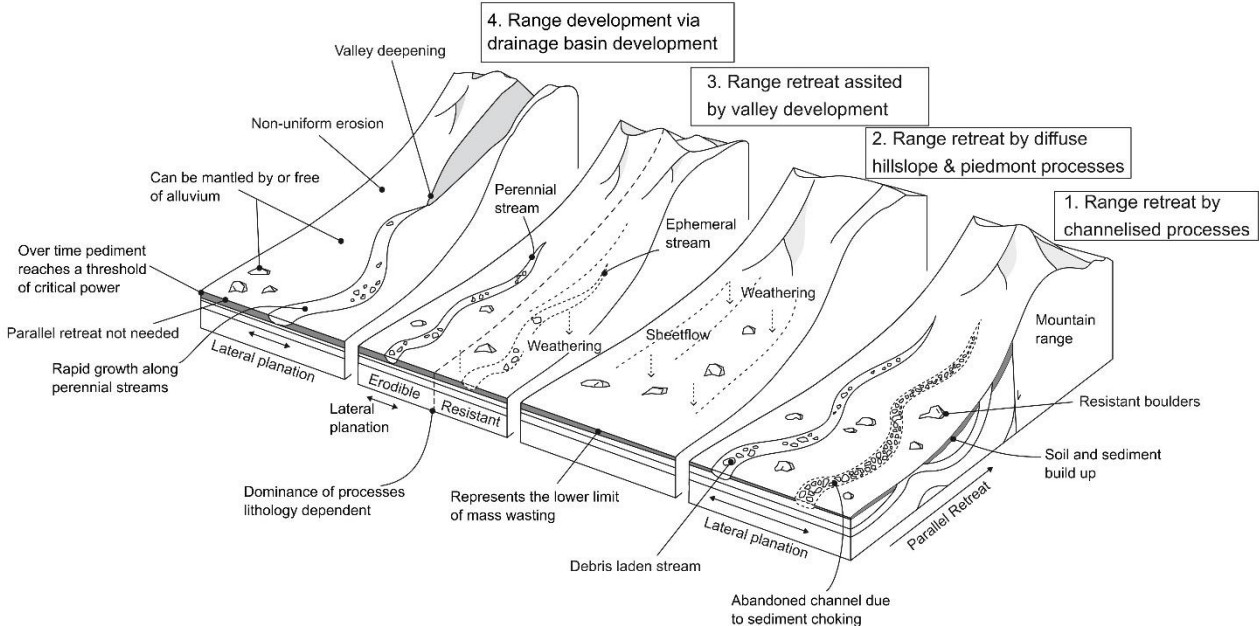

**Figure 1: Pediment evolution models showing the range of processes that can shape pediments; 1) Range retreat by channelised processes adapted from Gilbert, (1877), Paige (1912) and Howard (1942); 2) Range retreat by diffuse hillslope and piedmont processes adapted from Lawson (1915), Rich (1935), Kesel (1977), Bourne and Twidale (1998) and Dauteuil et al. (2015); 3) Range retreat assisted by valley development adapted from Bryan (1923) and Sharp (1940) and; 4) Range development via drainage basin development adapted from Lustig (1969) and Parsons and Abrahams (1984).**

The geomorphology of southern Africa has long intrigued earth scientists (Rogers, 1903; Davis, 1906; Dixey, 1944; King, 1948, 1949, 1953). Fundamental questions related to long-term landscape development remain contentious, such as the mechanisms and timing of surface uplift (e.g., Gallagher and Brown, 1999, Brown et al., 2002, Tinker et al., 2008b, Kouvnov et al., 2009, Decker et al., 2013; Wildman et al. 2015; Wildman et al. 2017; Stanley et al. 2021) and the chronological framework of the main phases of landscape development (Du Toit, 1937, 1954; King, 1951; Burke, 1996; Partridge, 1998; Brown et al., 2002; Doucouré and de Wit, 2003; de Wit, 2007; Kounov et al., 2015). In-situ produced cosmogenic nuclides can offer key information to unravel questions related to landscape development and evolution and have been applied to ancient landforms within southern Africa (Fleming et al. 1999; Cockburn et al., 2000; Bierman and Caffee, 2001; van der Wateren and Dunai, 2001; Kounov et al., 2007; Codilean et al., 2008; Dirks et al., 2010; Decker et al., 2011; Erlanger et al., 2012; Chadwick et al., 2013; Decker et al., 2013). However, studies based on in-situ produced cosmogenic studies, in the region south of the Great Escarpment are sparse (e.g., Scharf et al., 2013; Bierman et al., 2014; Kounov et al., 2015).



Pediments or erosional surfaces have been investigated in South Africa since the 1950's (King, 1953; King 1963; Partridge
and Maud, 1987), and have denudation rates that are an order of magnitude lower than those in other landforms within southern
Africa (van der Wateren and Dunai, 2001; Bierman et al., 2014; Kounov et al., 2015; Fig. 2). The pediment surfaces were
inferred as being early Cenozoic to Jurassic in age by King (1963). Large scale erosional features are also a feature of the
wider African continent, and extensive research has been undertaken to understand mantle dynamics associated with plateau
formation (e.g., Braun et al., 2014; Dauteuil et al., 2015; Guillocheau et al., 2015; Guillocheau et al., 2018). In this paper, we
present new isotopic data from pediment landforms in southern South Africa. The main aim of the paper is to constrain
landscape development using in-situ produced $^{10}$Be isotopes and to establish denudation rates and landform exposure ages.
The objectives of the paper are to: 1) assess the formative process associated with pediment evolution; 2) assess the cosmogenic
data within a wider geomorphic and geologic framework in order to test the performance of cosmogenic dating in a geomorphic
setting with very low denudation rates; and 3) discuss the implications for the wider landscape development of southern South
Africa.
**2 Regional Setting**
**2.1 Geological setting**
In the area of study of Western Cape, Southern Africa, the geology is dominated by strata of the Cape and Karoo Supergroups
(Fig. 2), which are composed of various sandstone, siltstone and mudstone successions. Both supergroups have been
metamorphosed, and the Karoo Supergroup has igneous intrusions. Tectonic shortening of Cape and Karoo Supergroups have
resulted in with E-W trending folds that decrease in amplitude northward and form the backbone of the exhumed Cape Fold
Belt (CFB)(Paton, 2006; Tinker et al., 2008b; Scharf et al., 2013; Spikings et al., 2015). During the Mesozoic, the rifting of
Gondwana initiated large-scale denudation across southern Africa. Using apatite fission track analyses of outcrop and borehole
samples, Tinker et al.(2008a) concluded that the southern Cape escarpment and coastal plain underwent 3.3 to 4.5 km of
denudation since the mid-late Cretaceous and potentially 1.5 to 4 km within the early Cretaceous, using a thermal gradient of
~20°C/km. Wildman et al. (2015) processed 75 apatite fission track and 8 zircon fission track data from outcrop and boreholes
across the southwestern cape of South Africa (from coast to the escarpment). Using a thermal history model of 22°C/km, they
obtained an average of 4.5 km of denudation in the Mesozoic. However, the estimates range between 2.2 and 8.8 km of
denudation using the upper and lower ranges of the geothermal gradient and possible thermal histories bounded by 95%
significance intervals, which provides uncertainty on the inferred model. Richardson et al. (2017) used reconstructed geological
cross sections and drainage reconstruction to model up to 4-11 km of denudation.





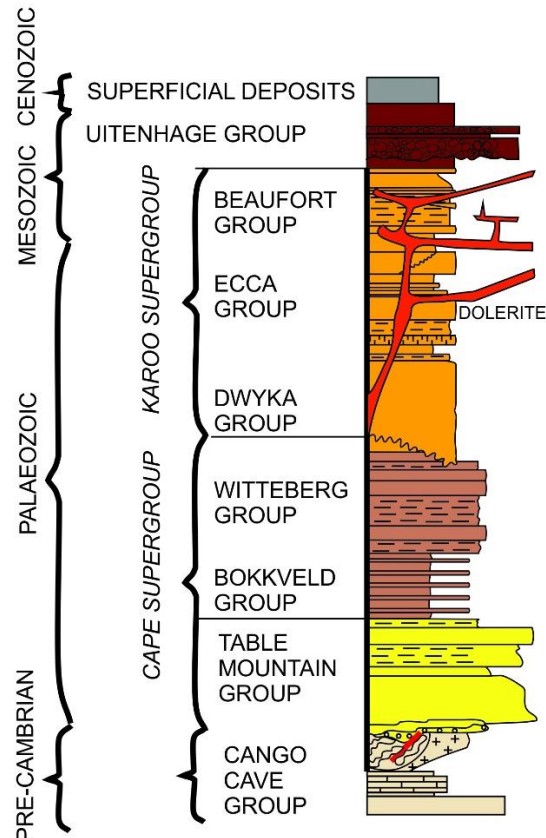

**Figure 2: Stratigraphic chart showing the major lithostratigraphic units of South Africa.**

The mechanisms of regional uplift since the Mesozoic, related to the anomalous height of southern Africa, are contentious; with landscape evolution either associated to mantle plumes (Nyblade and Robinson, 1994, Ebinger and Sleep, 1998) or to plate tectonics, with uplift along flexures (Moore et al., 2009) and epeirogenic uplift (Brown et al., 1990). Furthermore, the occurrence and timing of later Cenozoic uplift is disputed (e.g., Brown et al., 2002; van der Beek et al., 2002). Burke (1996) proposed that the most recent uplift phase occurred ~30 Ma ago due to a thermal anomaly. Green et al. (2016) also argued for Cenozoic uplift within southern South Africa that caused localised incision of the Gouritz River into the Swartberg mountain range. However, Partridge and Maud (1987) argued for two phases of uplift during the Neogene, with a phase around 18 Ma and a more recent phase at 2.58 Ma.

Figure 3 provides an overview of published geochronological studies in southern South Africa that used either apatite (U-Th)/He and apatite fission track analysis to document landscape denudation from the Cretaceous to modern day, or in-situ produced cosmogenic radionuclides ($^{26}$Al, $^{10}$Be, $^{3}$He, $^{21}$Ne) to date landforms. Apatite (U-Th)/He and fission track data (Fig. 3) indicate high rates of denudation (up to 175 m My$^{-1}$, Tinker et al., 2008b) with respect to the present day rates, towards the



end of the Lower Cretaceous (100– 80 Ma) that decreased to up to 95 m My$^{-1}$ by the late Cretaceous (90– 70 Ma; Brown et
al., 2002). Flowers and Schoene (2010) report negligible erosion since the Cretaceous, with rates as low as 5 m My-1 by the
late Eocene (36 My; Cockburn et al., 2000). Cosmogenic studies support low erosion rates within southern South Africa since
the start of the Cenozoic (Fig 3; Fleming et al., 1999; Cockburn et al., 2000; Bierman and Caffee, 2001; van der Wateren and
Dunai, 2001; Kounov et al., 2007; Codilean et al., 2008; Dirks et al., 2012; Decker et al., 2011; Erlanger et al., 2012; Chadwick
et al., 2013; Decker et al., 2013; Scharf et al., 2013; Bierman et al., 2014; Kounov et al., 2015). The majority of landforms are
eroding very slowly, with mean denudation rates ranging between 9.4 m My$^{-1}$ for the escarpment faces to 0.85 m My-1 for
pediments (Fig. 3), although one reported retreat rate of 62.3 m My$^{-1}$ have been measured for escarpment face retreat (Fleming
et al., 1999). In contrast, the Great Escarpment in the South African interior has higher fluvial incision rates than southern
South Africa: cosmogenic 3He channel bed denudation rates range between 14 and 255 m My$^{-1}$ and valley side and valley top
denudation rates range between 11 to 50 m My$^{-1}$ for the Klip and Mooi Rivers and Schoonspruit, tributaries of the Orange
River (Keen-Zebert et al., 2016).

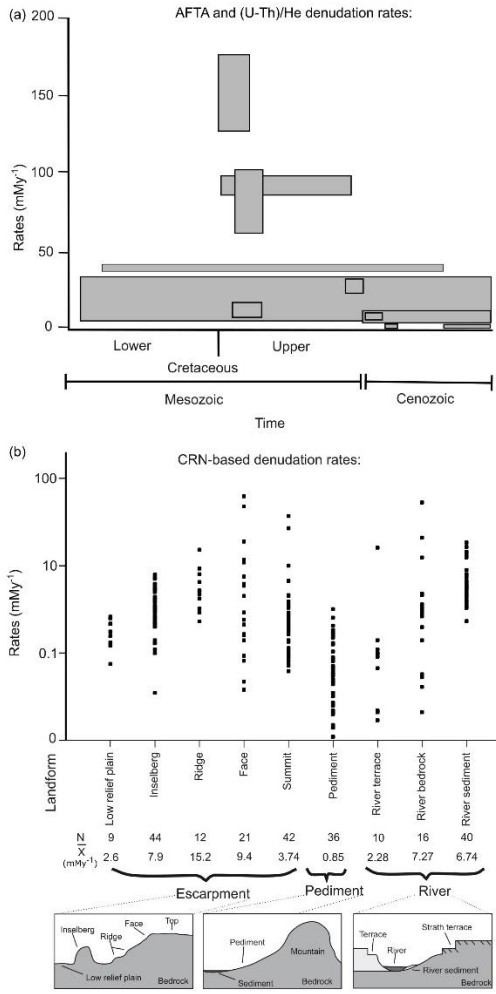

**Figure 3: Published exhumation and denudation rates for southern Africa. A) Apatite fission track and (U-Th)/He data show large variation in exhumation rates since the Cretaceous, and include data from Gallagher and Brown, 1999; Cockburn et al. 2000; Brown et al. 2002; Tinker et al. 2008b; Kounov et al. 2009 and; Flowers and Schoene, 2010. B) In-situ produced cosmogenic (10Be, 26Al, 21Ne and 3He) nuclide-derived denudation rates for escarpment, pediment and fluvial landforms. Cosmogenic data is from the following sources; Flemming et al. 1999; Cockburn et al. 2000; Bierman and Caffee, 2001; van der Wateren and Dunai, 2001; Kounov et al. 2007; Codilean et al. 2008; Dirks et al. 2012; Decker et al. 2011; Erlanger et al. 2012; Chadwick et al. 2013; Decker et al. 2013; Scharf et al. 2013; Bierman et al. 2014; and Kounov et al. 2015.**

Southern South Africa, below the Great Escarpment, is currently tectonically quiescent with only minor Quaternary-active faults (Bierman et al., 2014) and low denudation and sediment production rates (Kounov et al., 2007; Scharf et al. 2013). Minimum exposure ages for pediments range from 0.29 +/- 0.02 Ma (Bierman et al., 2014) to 5.18 +/- 0.18 Ma (Van der Wateren and Dunai, 2001) with a mean minimum exposure age of 1.87 Ma (Pleistocene, van der Wateren and Dunai, 2001; Bierman et al., 2014; Kounov et al., 2015).






The climate of southern South Africa has gradually moved towards more arid conditions since the Cretaceous (Partridge, 1997;
van Niekerk et al., 1999) with an abrupt change from humid/tropical to arid conditions at the end of the Cretaceous (Partridge
and Maud, 2000) as shown by silcrete formation and saline soils (Partridge and Maud, 1987). Although there is general
agreement about the overall aridification trend since the Cretaceous, several authors have argued that wetter phases occurred
from 65 – 30 Ma (Burke, 1996), or that the arid phase started as late as 18 Ma (Partridge and Maud, 1987). The present day
climate of the Western Cape is primarily semi-arid (Dean et al., 1995), while the coastal region has a Mediterranean type
climate (Midgley et al., 2003).
**2.2 Sample Sites**
The sampling sites are located within the large antecedent Gouritz catchment (Fig. 4), where morphometric analysis has
identified the presence of flat surfaces or pediments that carry a thin sedimentary cover, hereafter called alluviated pediments
(<1m) (Richardson et al., 2016). The alluviated pediments grade away from the Cape Fold Belt (CFB) into adjacent alluvial
plains, and samples were collected from pediments on the northern flank of the Swartberg and Witteberg Mountains (CFB)
around Laingsburg, Floriskraal, Leeuwgat, and Prince Albert (Fig. 4a). Samples were taken from five deeply dissected
alluviated pediments ranging in surface area between < 1 to 20 km$^2$ and displaying slope angles below 10°, with most of the
slopes below 4° (Fig. 5).





**Figure 4: (a) Pediment locations, the inset shows the location of the Gouritz catchment within South Africa, where CT – Cape Town, LB – Laingsburg; GM – Gouritzmond and the red polygon is the location of the Cape Fold Belt (CFB); (b) underlying geology below the pediments and; (c) pediment elevations (in m a.s.l.) as shown by elevation bins categorised by natural breaks in the elevation data.**



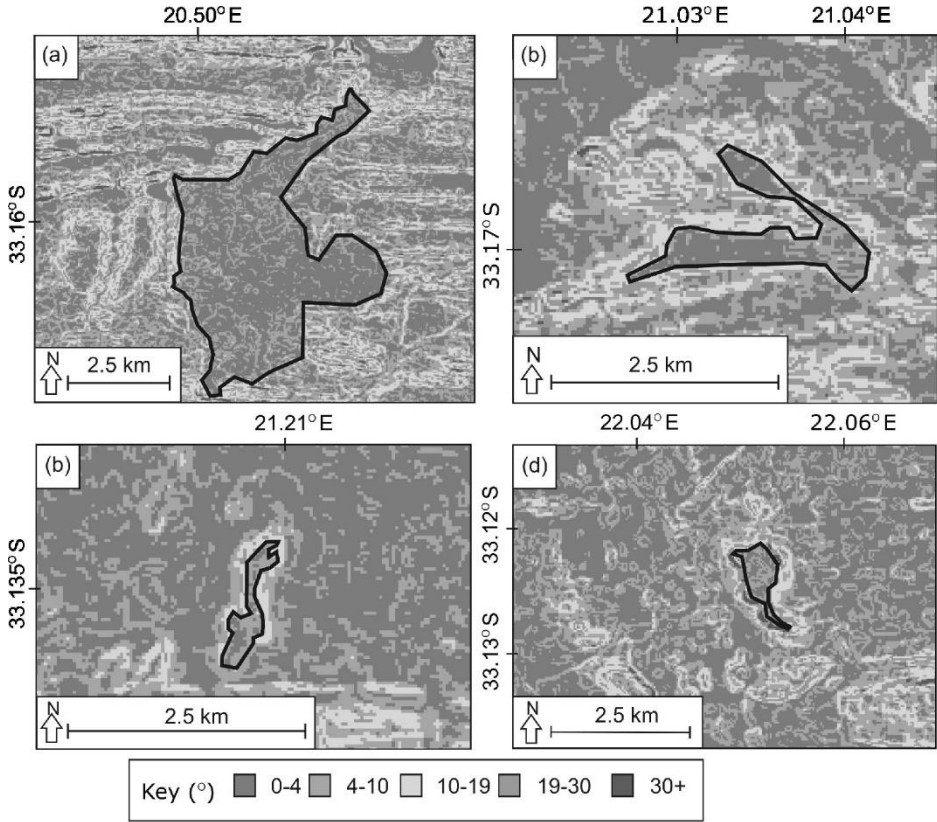

**Figure 5: Pediment slope data (with slope given in °); (a) Laingsburg; (b) Floriskraal; (c) Leeuwgat and; (e) Prince Albert. For pediment locations please see Figure 4.**

The alluviated pediments are composed of unconsolidated, poorly-sorted gravel to boulder material in a matrix of sand (Fig. 6) that unconformably overlie folded rocks of the Karoo Supergroup (Fig. 3b). Some pediments are capped by silcrete, calcrete or ferricrete (Helgren and Butzer, 1977; Summerfield, 1983; Marker and Holmes, 1999; Partridge, 1999; Partridge and Maud, 2000; Marker et al., 2002). Ferricrete is dominant on the Laingsburg pediment. The silcrete is assigned to the Grahamstown Formation (Fig. 4b) that has poor age control (Mountain, 1980; Summerfield, 1983) due to the lack of formal identification of the extent of the silcretes. Electron spin resonance ages for two silcrete caps in the Kleine Karoo were dated at 7.3 and 9.4 Ma (Hagedorn, 1988).





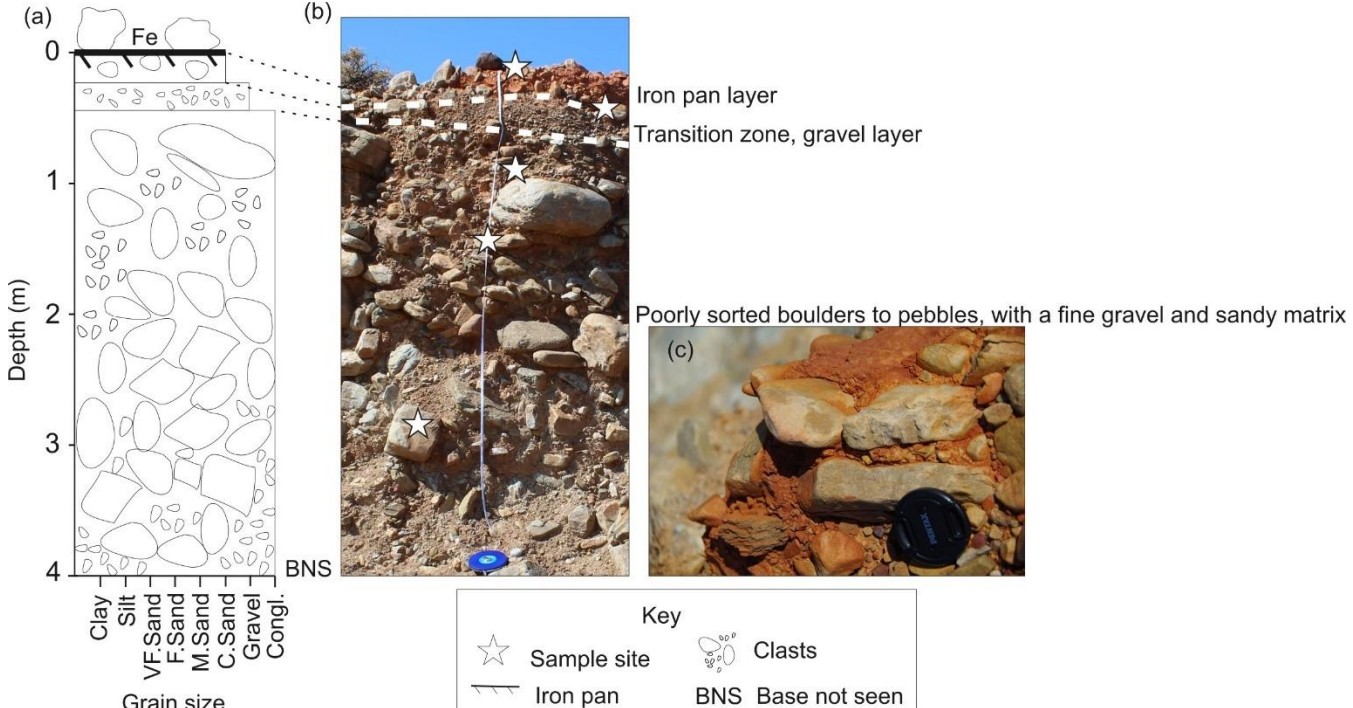

193

**Figure 6: (a) Sedimentary log of the Laingsburg pediment showing the unsorted boulders (dominantly quartzite) to**
**gravel size material; (b) photograph of the pediment and where the depth profile clasts were taken; (c) iron-rich**
**palaeosol layer.**

## 3. Methodology

### 3.1 Cosmogenic radionuclide dating

Two types of samples were collected for CRN analyses in 2014: five rock samples from alluviated pediment surfaces and

clasts from one depth profile in the Laingsburg pediment (Fig. 7, Table 1). Quartzite boulders from the Table Mountain

Group (Cape Supergroup) that were sampled at the surface of the pediments have a >1m diameter along their longest axis.

For the depth profile in the pediment, quartzite clasts (>25 cm diameter) were taken at the following depths (cm) below

ground level: 0, 30, 85, 150, 255 (Table 1).





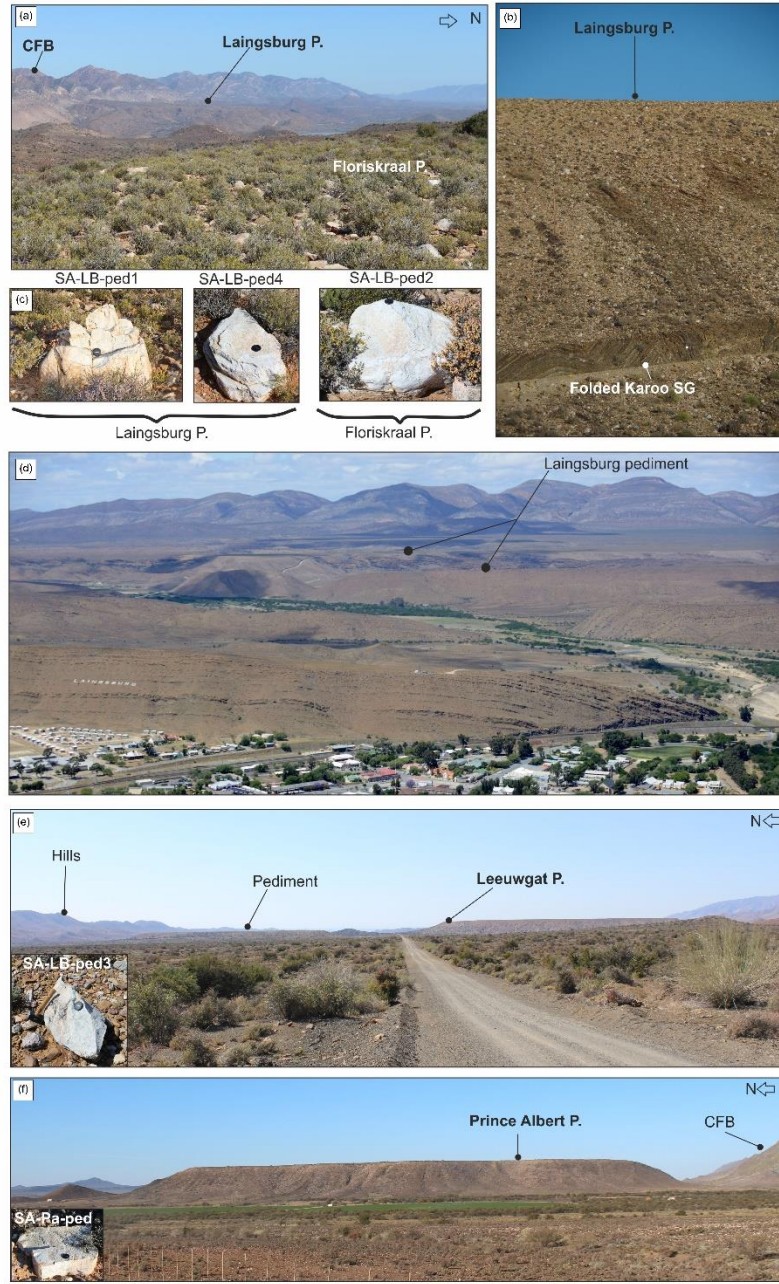

204

**Figure 7: Sample sites; (a) Laingsburg pediment from the Floriskraal pediment; (b) Laingsburg pediment and contact with underlying folded Karoo Supergroup (SG) strata; (c) Boulder samples from Laingsburg and Floriskraal pediments; (d) large-scale picture of the Laingsburg pediment; (e) Leeuwgat pediment and boulder sample (inset); (f) Prince Albert and boulder sample (inset). The figure also shows the dissection of the pediments by small river catchments and how decoupled the Floriskraal and Prince Albert pediments are from the Cape Fold Belt.**



**Table 1: Site-specific information of the sampling sites for cosmogenic radionuclide analysis. All samples are taken from quartzite boulders, that were sampled either on the surface of the pediment (sample type = surf) or at depth (sample type = depth). The density of the sample or overburden (for depth samples) has been determined based on published density data of quartzite boulders and depth profiles in pediments by respectively Scharf et al. (2013) and Kounov et al. (2015).**

| Sample ID | Sample type | Name | Latitude (°S) | Longitude (°E) | Elevation (m) | Density (g/cm³) | Topographic Shielding | Cover correction |
|---|---|---|---|---|---|---|---|---|
| SA-PA_ped | Surf | Prince Albert | 33.203 | 22.082 | 703 | 2.7 | 1.00 | NA |
| SA-LB_ped1 | Surf | Laingsburg | 33.246 | 20.872 | 764 | 2.7 | 1.00 | NA |
| SA-LB_ped2 | Surf | Floriskraal | 33.285 | 21.050 | 706 | 2.7 | 1.00 | NA |
| SA-LB_ped3 | Surf | Leeuwgat | 33.221 | 21.347 | 691 | 2.7 | 1.00 | NA |
| SA-LB_ped4 | Surf | Laingsburg | 33.261 | 20.854 | 791 | 2.7 | 1.00 | NA |
| SA-LB_DP0 | Depth | Laingsburg | 33.256 | 20.851 | 779 | 1.6 | 0.99 | NA |
| SA-LB_DP30 | Depth | Laingsburg | 33.256 | 20.851 | 776 | 1.6 | 0.99 | 0.79 |
| SA-LB_DP85 | Depth | Laingsburg | 33.256 | 20.851 | 776 | 1.6 | 0.99 | 0.54 |
| SA-LB_DP150 | Depth | Laingsburg | 33.256 | 20.851 | 776 | 1.6 | 0.99 | 0.37 |
| SA-LB_DP255 | Depth | Laingsburg | 33.256 | 20.851 | 776 | 1.6 | 0.99 | 0.23 |

The samples were processed for in-situ cosmogenic $^{10}$Be following standard methods as described in von Blanckenburg (2004) and Vanacker et al. (2007). Rock samples were crushed, sieved and rock fragments of 250 to 500 µm diameter were selected for further lab processing. Quartz minerals were extracted by chemical leaching with a low concentration of acids (HCl, HNO$_3$, and HF) in an overhead shaker. Purified quartz samples were then leached with 24% HF for 1h to remove meteoric $^{10}$Be, followed by spiking the sample with 150 µg of $^9$Be and total decomposition in concentrated HF. The Beryllium in solution was extracted by ion exchange chromatography as described in von Blanckenburg et al. (1996). The $^{10}$Be/$^9$Be ratios were measured using accelerator mass spectrometer on the 500 kV Tandy facility at ETH Zürich (Christl et al., 2013). Measured $^{10}$Be/$^9$Be ratios were normalised to the ETH in-house secondary standard S2007N with a nominal ratio of 28.1×10$^{-12}$ (Kubik and Christl, 2010), which is in agreement with a $^{10}$Be half-life of 1.387 Ma (Chmeleff et al., 2010). Sample ratios were blank corrected (7.54 ± 9.67 × 10$^{-15}$) and the analytical uncertainties on the $^{10}$Be/$^9$Be ratios of blanks and samples were then propagated into the 1σ analytical uncertainty for the $^{10}$Be concentrations (Table 2 and 3). Production rates were scaled following Dunai (2000) with a sea level high-latitude production rate of 4.28 atoms g$_{qtz}^{-1}$ yr$^{-1}$. The bulk density was set to 2.7 g cm$^{-3}$ for samples from quartzite boulders following Scharf et al. (2013), and to 1.6 g cm$^{-3}$ for the overburden of the depth samples following earlier work on depth profiles in the Western Cape by Kounov et al. (2015). The concentrations were corrected for topographic shielding using the procedure described in Norton and Vanacker (2009).



**Table 2 : Cosmogenic nuclide data for depth profile in Laingsburg. The reported $^{10}$Be concentrations are corrected for procedural blanks, using a value of $7.54 \pm 9.67 \times 10^{-15}$, and the 1σ uncertainty estimates contain analytical errors from AMS measurement and blank error propagation.**

| Sample ID | Depth (cm) | $^{10}$Be concentration ($\pm 1\sigma$), ($\times 10^6$ at/g$_{qtz}$) |
|---|---|---|
| SA-LB_DP0 | 0 | $5.460 \pm 0.106$ |
| SA-LB_DP30 | 30 | $1.196 \pm 0.111$ |
| SA-LB_DP85 | 85 | $0.893 \pm 0.036$ |
| SA-LB_DP150 | 150 | $0.376 \pm 0.016$ |
| SA-LB_DP255 | 255 | $0.133 \pm 0.015$ |

**Table 3: Cosmogenic nuclide data for surface samples from pediments. The reported $^{10}$Be concentrations are corrected for procedural blanks, using a value of $7.54 \pm 9.67 \times 10^{-15}$, and the 1σ uncertainty estimates contain analytical errors from AMS measurement and blank error propagation. Maximum denudation rates and minimum durations of surface exposure were calculated using the CosmoCalc add-in for Excel (Vermeesch, 2007). For the surface exposure ages, we assumed (1) no erosion or burial since exposure, and (2) a maximum steady erosion rate of 0.3 m My$^{-1}$.**

| Sample ID | Location | $^{10}$Be concentration ($\times 10^6$ at/g$_{qtz}$) ($\pm 1\sigma$) | $^{10}$Be denudation rate (m My$^{-1}$) ($\pm 1\sigma$) | Minimal exposure age (My) ($\pm 1\sigma$) | |
|---|---|---|---|---|---|
| | | | | No erosion or deposition | Erosion rate of 0.30 m My$^{-1}$ |
| SA-PA_ped | Prince Albert | $2.834 \pm 0.055$ | $0.954 \pm 0.025$ | $0.569 \pm 0.010$ | $0.678 \pm 0.010$ |
| SA-LB_ped1 | Laingsburg | $5.199 \pm 0.096$ | $0.408 \pm 0.013$ | $1.131 \pm 0.016$ | $1.964 \pm 0.016$ |
| SA-LB_ped2 | Floriskraal | $5.148 \pm 0.095$ | $0.383 \pm 0.013$ | $1.189 \pm 0.016$ | $2.220 \pm 0.016$ |
| SA-LB_ped3 | Leeuwgat | $5.641 \pm 0.103$ | $0.315 \pm 0.011$ | $1.377 \pm 0.018$ | $4.462 \pm 0.018$ |
| SA-LB_ped4 | Laingsburg | $4.252 \pm 0.067$ | $0.587 \pm 0.014$ | $0.848 \pm 0.011$ | $1.164 \pm 0.010$ |
| SA-LB_DP0 | Laingsburg | $5.460 \pm 0.106$ | $0.373 \pm 0.013$ | $1.210 \pm 0.018$ | $2.333 \pm 0.018$ |





For the derivation of the minimum durations of exposure (Table 3), we used two different scenarios: a hypothetical case
assuming no erosion or burial since exposure, and a second case assuming steady erosion of the pediment surface of 0.3m My⁻
¹ following Bierman et al. (2014). The CosmoCalc method, version 3.0 (Vermeesch, 2007) was employed to calculate
maximum denudation rates and minimum surface exposure ages from the $^{10}$Be concentrations of the surface samples (Table
3). The surface exposure ages are *minimum estimates* as isotopic steady state can be reached for old material.
In addition, we use a concentration depth profiling approach to better constrain the exposure and denudation of the Laingsburg
area pediment. The accumulation of $^{10}$Be, $N_{total}$ (z,t), in the eroding surface of the pediments can be described as:
$$N(z,t) = N_{inh}e^{-\lambda t} + \sum_i \frac{P_i(z)}{\lambda + \frac{\rho E}{\Lambda_i}} e^{-\rho(z_0 - Et)/\Lambda_i} \left(1 - e^{-(\lambda + \frac{\rho E}{\Lambda_i})t}\right) \qquad \text{Eq.1}$$
where $E$ is expressed in cm/yr (m/Myr × $10^4$), $t$ [yr] is the exposure age, $\lambda$ [1/yr] the nuclide decay constant ($\lambda$ = ln 2 /$t_{1/2}$), $z_0$
(cm) the initial shielding depth ($z_0 = E \times t$), $\rho$ [g/cm³] the density of the overlying material, and $\Lambda_i$ [g/cm²] the attenuation
length. The production rate, $P_i(z)$ [atoms/g/yr], is a function of the depth, $z$ [cm], below the surface. The subscript 'i' indicates
the different production pathways of $^{10}$Be via spallation, muon capture and fast muons following Dunai (2010). In this study,
the relative spallogenic and muogenic production rates are based on the empirical muogenic-to-spallogenic production ratios
established by Braucher et al. (2011), using a fast muon relative production rate at SLHL of 0.87% and slow muon relative
production rate at SLHL of 0.27%.The attenuation length was set to 160, 1500 and 4320 g cm⁻² for the production by,
respectively, neutrons, negative muons and fast muons (Braucher et al., 2011). The depth profile is then solved numerically,
based on a chi-squared model fitting between the observed (Table 2) and simulated $^{10}$Be concentrations at different depths.
**3.2 Morphometric Analysis**
Aster 30m data was used to build a DEM of the study area in ArcGIS 10.1. The DEM was re-projected into WGS 1984 world
Mercator co-coordinates and filled using the hydrology toolbox. The drainage was extracted using an upstream contributing
area of 3.35 km², and both ephemeral and perennial streams were delineated (e.g., Abadelkaarem et al., 2012; Ghosh et al.,
2014). Dissected pediments were derived using a method adapted from Bellin et al. (2014). The previous grading from the
mountain front was reconstructed for each pediment in ArcGIS (Fig. 8). This surface was then placed into ArcScene 10.1, with
the difference between the reconstructed surface and the current topography (using the DEM) providing a minimum volume
of material removed after pediment formation. A similar approach was applied to derive bulk erosion volumes for the small
sub-catchments that back the pediment surfaces in the CFB. The bulk erosion is likely to be a minimum estimate of the total
rock volume removed by erosion, as interfluve erosion might have occurred (Bellin et al., 2014; Brocklehurst and Whipple,
2002). Eroded volumes were then converted to lithological thickness using the method of Aguilar et al. (2011).



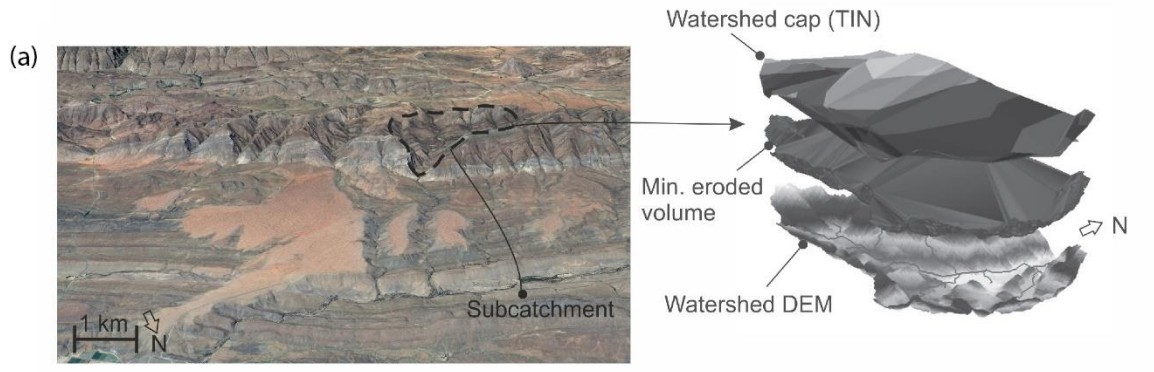

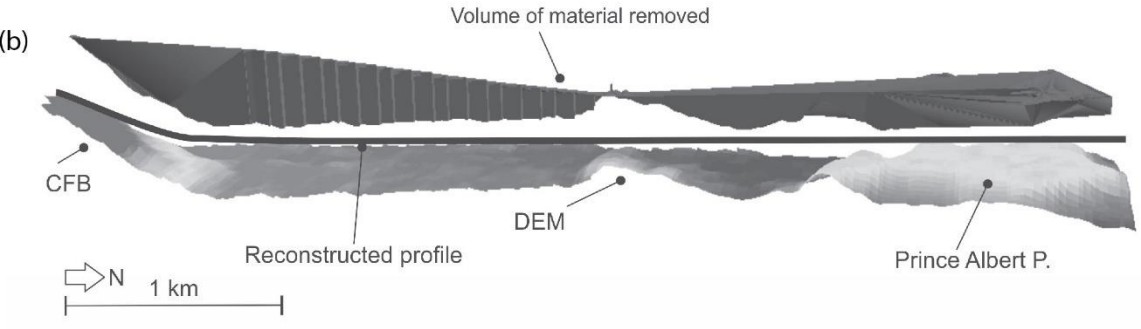

**Figure 8: Examples of (a) bulk eroded volumes from subcatchments and (b) cross section of the Prince Albert pediment showing the method used in ArcGIS for the volume of material removed around the pediment surface.**

## 4. Results

### 4.1 Alluviated pediment composition

The contact with the underlying bedrock (e.g., Dwyka Group) is erosional and undulating, it is not a smooth planation contact. The alluviated pediments are composed of poorly sorted boulders to pebbles, with a matrix of sandy gravel. The clasts are predominantly quartzites (Table Mountain Group); however smaller clasts of Dwyka Group lithologies are present. Towards the top of the profile there is a small transition zone of gravel, which is capped by an iron crust (Fig. 6). There is no indication of fluvial activity (i.e., imbrication). There is no grading or sediment clast size variation throughout the profile, and the clasts range from sub-rounded to sub-angular.





## 4.2 Cosmogenic nuclides

The surface lowering rates (Table 3) calculated for the boulders sampled on the pediment surface show very low maximum denudation rates, which range from 0.315 to 0.954 m My$^{-1}$. The Laingsburg area alluviated pediment has higher rates of surface lowering closer to the CFB, with denudation rates decreasing towards the proximal part of the pediment as shown by the boulder samples. The alluviated pediment in the Prince Albert area has the highest rate of maximum surface lowering (0.954 m My$^{-1}$), which is an order of magnitude higher than the average surface lowering rate of the other studied alluviated pediments. The minimum exposure ages assuming no erosion or burial (Table 3) indicate that the alluviated pediments are long-lived, with *minimum* surface exposure ages between 0.569 and 1.377 My (Pleistocene). The Prince Albert area alluviated pediment has the youngest *minimum* exposure age of 0.569 My, the Laingsburg area pediment has variable minimum exposure ages from 0.848 to 1.131 My. Over this timeframe, the assumption of no erosion or deposition is an unlikely scenario. Assuming low erosion rates of 0.3 m My$^{-1}$ the pediment minimum exposure ages increase substantially for the older surfaces, with minimum ages ranging from 0.678 to 4.462 My (Table 3).

The $^{10}$Be concentration depth profile provides more insights in the denudation process of the pediments. First, the uppermost sample of the Laingsburg depth profile has a $^{10}$Be concentration that is in line with the concentrations that are measured in boulders sampled at the Laingsburg, Floriskraal and Leeuwgat alluviated pediments, and is markedly higher than the concentration measured at the Prince Albert alluviated pediment (Fig. 9a, Table 3). Second, there is a large discrepancy in the $^{10}$Be concentrations between the uppermost sample and the four samples taken at depth in the profile (Table 2). The $4.265 \times 10^6$ at./g difference in $^{10}$Be concentrations over a 30 cm depth increment cannot be explained by steady erosion of the pediment after exposure (Fig. 9b). It suggests that deflation of ~110 cm of fine-grained material at the surface of the pediments has resulted in a pavement of old boulders at the top of a slowly eroding surface (Fig. 10).





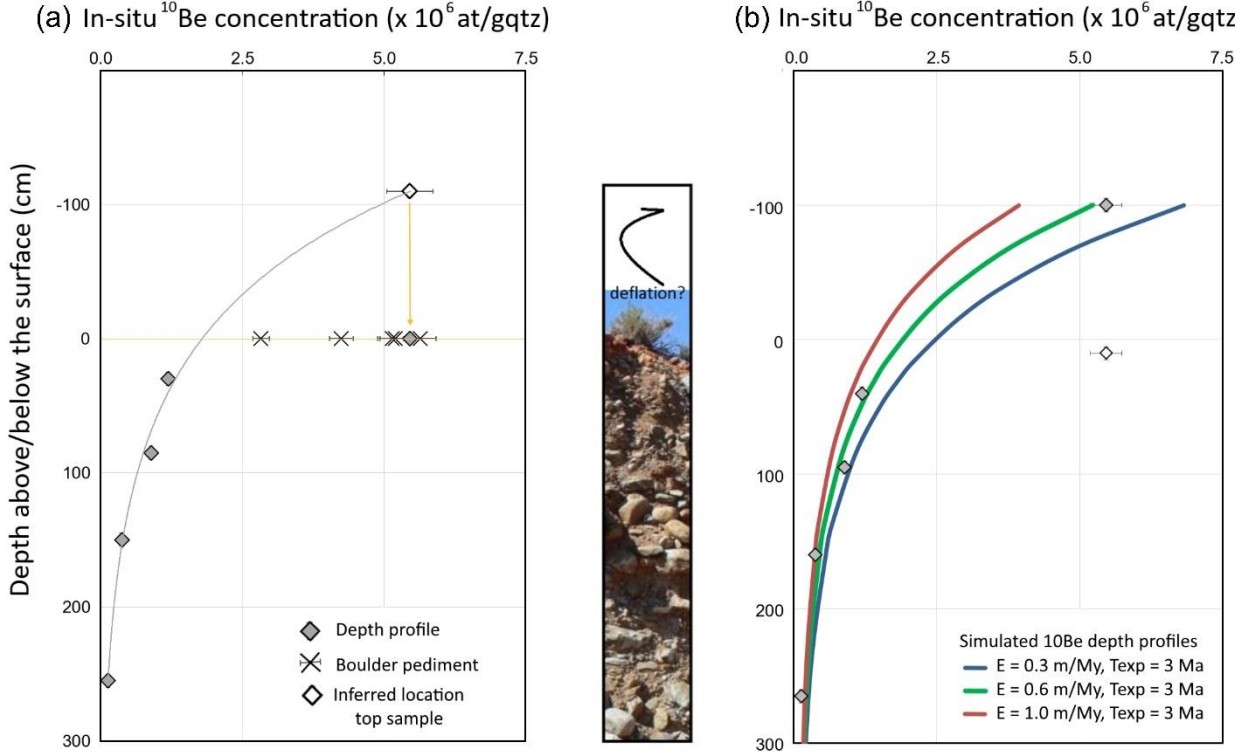


**Figure 9: Depth profile results of the Laingsburg pediment. (a) showing depth profile data and (b) showing erosion**
**rate scenarios.**




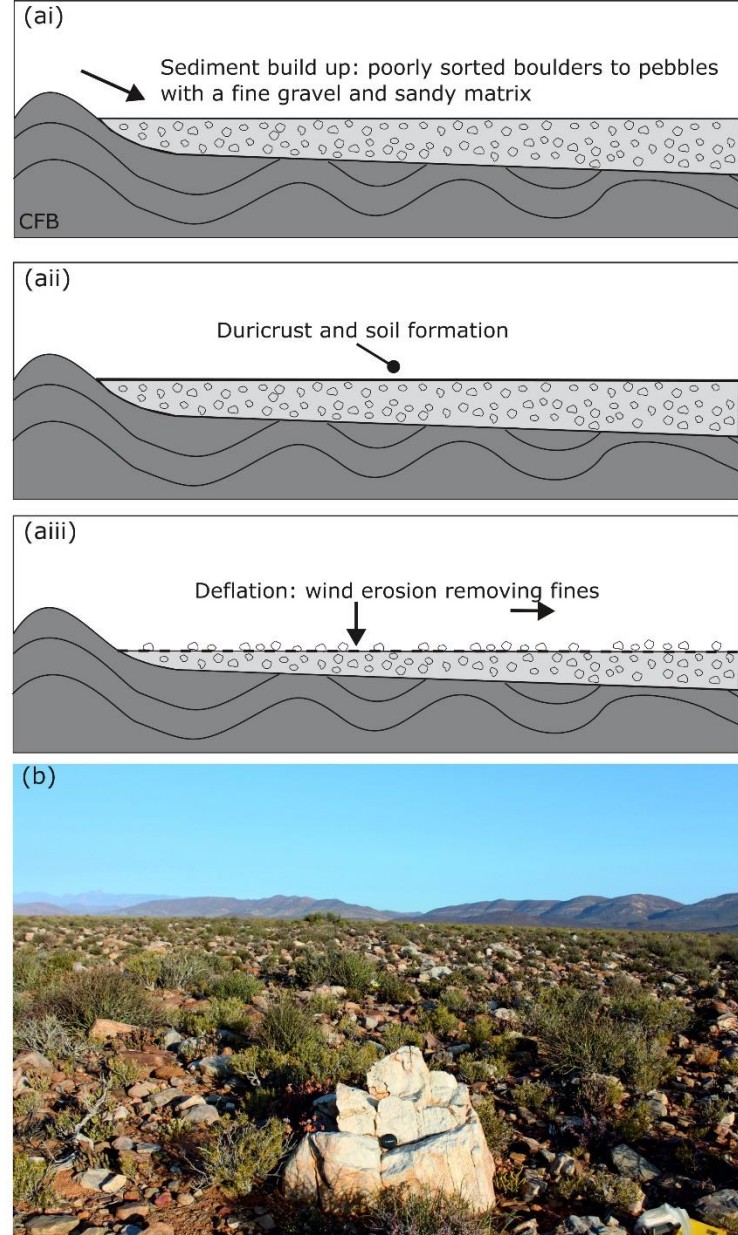


**Figure 10: (a) Process of deflation and (b) Evidence of deflation: concentrations of boulders and pebbles on top of the Laingsburg Pediment.**




When taking ablation of the upper ~110cm of the profile into account, the [10]Be concentration depth profile of the Laingsburg
pediment can be simulated by forward modelling (Vandermaelen et al., 2022) using minimum age constraints from the surface
samples and information on the density of the overlying material from Kounov et al. (2015). The most likely denudation rate
of the pediment is ~0.6 m/My (Figure 9b), which is similar to the median erosion rate for South African pediment surfaces
reported by Bierman et al (2014). Even at this low surface lowering rate, the [10]Be concentrations approach isotopic steady state
when the time of exposure exceeds 3 Ma, so that the age information derived from the depth profile only provides a *minimum*
*exposure age*.
**4.3 Elevations and grading of pediment**
Figure 4c shows the pediment heights as classified by the Jenks natural break scheme (De Smith and Goodchild, 2007). The
alluviated pediments at Laingsburg and Floriskraal have elevations within the same class (714 – 870 m), and the Leeuwgat
and Prince Albert area alluviated pediments share the same elevation class (617 – 713 m). The Laingsburg area alluviated
pediment appears to have an aspect of slope that grades not only away from the CFB but towards the modern Buffels River
location, which abuts the northern limit of the alluviated pediment (Fig. 11). This relationship is less clear on the Floriskraal
alluviated pediment, which is to the east of the Buffels River. The alluviated pediment at Leeuwgat, which sits between two
folds of the CFB, has no large trunk river nearby (~30 km from Dwyka River) and simply grades away from the CFB (Fig.
12A). The Prince Albert area pediment grades towards the Gamka River, although it is currently ~16 km from the Gamka
River (Fig. 12b). The fact that the alluviated pediments grade towards the present day trunk rivers but above their present day
elevation indicates that these rivers were active during the formation of the pediments and is discussed later.





**Figure 11: Grading of the Laingsburg pediment and related cross sections, which grade not only away from the Cape Fold Belt but towards the Buffels River.**



**Figure 12: Grading of the (a) Leeuwgat, which grades away from the Cape Fold Belt and (b) Prince Albert pediment, which grades towards the Gamka River.**

**4.4. Dissecting river planform**

The dissecting river planforms are shown in Fig. 13. critical points are highlighted that relate to sections where the rivers (i) have been deflected by the pediment surface, or (ii) have anomalous changes in orientation. Overall, the low order rivers (<4) that have dissected the pediments are strongly influenced by the folding within the CFB (Richardson et al., 2016). This is especially seen within the rivers that have dissected the Laingsburg pediment (Fig. 13a), where the linear river planform aligns





344    with the axis of a syncline. Where the rivers breach the folds it appears that the presence of alluviated pediments deflected the

345    river planforms; this relationship can also been seen at Floriskraal and Prince Albert area alluviated pediments (Fig. 13).

346

**Figure 13: Planforms of the dissecting rivers and Cape Fold Belt subcatchments; (a) Laingsburg; (b) Floriskraal; (c) Leeuwgat and; (d) Prince Albert. The circles highlight critical points related to deflection of the river planforms by the Cape Fold Belt or the pediment.**



## 4.5 Volume of material removed

Table 4 shows the bulk erosion rates related to dissection of the alluviated pediment post formation. Converting this to an equivalent lithological thickness (dividing the volume of material removed over the area; Aguilar et al., 2011), an average of 141.43 m has been eroded around the large Laingsburg area pediment (Fig. 11). The Prince Albert area pediment, has an average lithological thickness of 42.33 m removed. Leeuwgat has had the least amount of dissection, with 17.25 m eroded.

**Table 4: Minimum volume of material eroded by rivers incising the pediment surface, the equivalent rock thickness and the time taken for incision using the average maximum denudation rate of 10.16 m My$^{-1}$ from Scharf et al., 2013 and Kounov et al., 2015.**

| Location | Volume of material removed (km$^3$) | Equivalent average rock thickness (m) | Time for incision (Ma) |
|---|---|---|---|
| Laingsburg | 3.240 | 141.43 | 13. 92 |
| Floriskraal | 0.154 | 42.33 | 4.17 |
| Leeuwgat | 0.169 | 44.27 | 4.36 |
| Prince Albert | 0.012 | 17.25 | 1.70 |

Table 5 shows the volume of material eroded by rivers draining the sub-catchments in the CFB, which have dissected the alluviated pediments. The sub-catchments range in area from 4.9 – 310 km$^2$, and the volume of material removed ranges from 0.11 - 89 km$^3$, which is the equivalent of 21 - 286 m of lithological thickness. The alluviated pediments that are located further away from the CFB range have larger dissecting catchments associated with them. For example, the Laingsburg area alluviated pediment, which is backed by the CFB, has an average sub-catchment area of 14.37 km$^2$, whereas the Prince Albert area alluviated pediment is located ~ 2 km from the CFB and has an average sub-catchment area of 161.83 km$^2$. These sub-catchment areas are contributing to the incision of the pediments.



**Table 5: Minimum volume of material eroded by rivers draining the Cape Fold Belt sub-catchments, the equivalent rock thickness and the average time taken for incision using the average of the maximum denudation rate recorded from Scharf et al., 2013 and Kounov et al., 2015 of 10.16 m My$^{-1}$.**

| Location | Catchment | Area (km²) | Volume of material removed (km³) | Equivalent average rock thickness (m) | Time for incision (Ma) |
|---|---|---|---|---|---|
| Laingsburg | CFB 1 | 19.79 | 2.86 | 144.39 | 14.21 |
| | CFB 2 | 8.96 | 0.85 | 95.55 | 9.40 |
| Floriskraal | CFB 3 | 6.21 | 0.28 | 45.31 | 4.46 |
| | CFB 4 | 6.02 | 0.20 | 33.59 | 3.31 |
| Leeuwgat | CFB 5 | 73.80 | 7.55 | 102.25 | 10.06 |
| | CFB 6 | 4.91 | 0.11 | 21.64 | 2.13 |
| Prince Albert | CFB 7 | 310.75 | 89.01 | 286.44 | 28.19 |
| | CFB 8 | 12.92 | 0.23 | 17.79 | 1.75 |

## 5. Discussion

### 5.1 Pediment formation and characteristics

The pediments are underlain by folded strata of the Karoo and Cape Supergroups (sandstone, siltstone and mudstone), and backed by the resistant CFB quartzites (Fig. 4b). It has been argued that pediments form on all lithology types, however the more extensive pediments can be found above the least resistant material (Dohrenward and Parsons, 2009). There is no systematic variation in pediment characteristics that can be related to the underlying geology (Fig. 4b).

The pediments have formed by diffusive processes, dominated by slope processes in the first stages of development, causing the gradual retreat of the Cape Fold Belt and coeval formation of colluvial material and the weathering mantle, including an iron pan (Fig. 14). There is no evidence of fluvial activity, such as clast imbrication, depositional or erosional bedforms, or channel-forms (Fig. 6; *cf.* e.g., Gilbert, 1877; Sharp, 1940; Lustig, 1969). The iron pan layer is now at the surface of the pediment due to the removal of overlying material as a result of surface deflation by wind erosion, as shown by the cosmogenic data from the ¹⁰Be concentration depth profile (Figs. 9, 14). The pediments grade towards, but above, large trunk rivers of the Gouritz catchment (Figs. 11, 12), indicating that large transverse systems were active before pediment planation and colluvial build-up. The trunk rivers were also active during pediment formation, however they were probably less so, as shown by the build-up and preservation of material forming the pediments. This suggests that at the time of pediment formation there was deposition of colluvial material adjacent to large-scale sediment bypass via rivers, and formation of the pediment surfaces





because of erosion processes. The trunk rivers, active during the formation of the pediments represent an upper limit to the
extent of the pediments and the pediments should be regarded as individual landforms and not as an extensive regional 'surface'
within the study area (*cf.* King, 1948, 1953, 1955; Partridge and Maud, 1987).

**Figure 14: Sequence of events forming the pediments and boundary conditions; in which the folded Karoo**
**Supergroup strata was planned, hillslope processes caused the build-up of sediment, soil formation and duricrust**
**formation. The pediments were then dissected and fluvial processes dominate. In recent time, deflation processes**
**have dominated (Fig. 10).**





The distribution of the dissected pediments suggests that these are remnants of much more continuous local features (Fig. 12).
There has been a shift in the dominant process regime, from slope processes to fluvial processes, during the evolution of the
pediments as evidenced by the dissection of pediments by smaller rivers and the decoupling of the pediments from the CFB
sediment source area. The river planform has been primarily controlled by the orientation of tectonic folds. However, the
pediments could have also controlled the landscape evolution by deflecting the rivers, allowing the surfaces to be preserved.
It appears that the structural integrity of the pediment is not continuous across the entire pediment, and areas underlain by
cohesive material caused deflection of the dissecting rivers due to a higher resistance to erosion (Fig. 13). This could be a
function of the sedimentology (Fig. 6) of the pediment: the calibre of material; the extent of packing; or the presence and
thicknesses of the duricrust layer. Deflection of rivers has been shown to cause the formation of epigenetic gorges (Ouimet et
al., 2008). Furthermore, the pediments could have been preserved in these locations as rivers did not migrate laterally, which
could be due to variations in channel gradient. The pediments sit above the valley floor (current level of erosion) and are
fossilised landforms that represent a store of sediment that is mostly subject to weathering and deflation under current climatic
conditions (Fig. 10), with hillslope processes slowly supplying sediment to the nearby fluvial channels; however due to slow
runoff rates related to the arid climate, the transport is no longer effective.

**5.2 Implications of depth profile**

The CRN concentration depth profile (Table 2) indicates that the $^{10}$Be concentrations in the sedimentary sequence deviate from
a simple exponential concentration depth profile. The stronger than theoretically expected decrease in $^{10}$Be concentrations in
the upper 30 cm point to a complex post-depositional history of the alluviated pediment at Laingsburg. The deviation can be
explained by a first phase of low denudation rates (0.6 m/My) followed by a second phase of aeolian deflation of the surface
whereby finer material is preferentially removed.
Deflation has been reported for (semi-)arid environments during the Cenozoic (Binnie et al. 2020). The impact of deflation on
$^{10}$Be concentrations has been described for glacial outwash terraces (Hein et al. 2009; Darvill et al. 2015) where aeolian
deflation and bio- or cryoturbation caused previously buried cobbles to become exposed. It has also been recorded for
periglacial areas of central Europe where depth profiles indicate denudation rates of 40 to 80 mMy$^{-1}$ during the Quaternary
(Ruszkiczay-Rudiger et al. 2011). Binnie et al. (2020) showed that deflation on marine terraces in Northern Chile is the primary
cause for multimodal distributions of $^{10}$Be concentration depth profiles.
Although the climate in southern South Africa has become more arid since the Cenozoic, the impact of aeolian deflation on
$^{10}$Be concentrations of pediment surfaces has not yet been addressed. The results from the $^{10}$Be concentration depth profile
indicate that caution should be taken when collecting only surface samples from alluvial pediment surfaces: boulders
armouring the surface of alluvial pediments can be enriched in $^{10}$Be concentrations, compared to the sandy matrix, as they are
residual features. Based on the complex $^{10}$Be concentration depth profile in the Laingsburg pediment, CRN-based denudation





rates from boulders could underestimate recent phases of surface deflation. Further work is needed to understand if this
behaviour is apparent across other pediment surfaces in the area, and how common this feature is across other pediment
surfaces. Future work should include concentration depth profiles from other alluvial pediments to ascertain if surface deflation
is occurring, and to account for this process when establishing regional long-term denudation rates from CRN.

**5.3 Geomorphic, tectonic, climatic and stratigraphic considerations**

The cosmogenic data presented in Table 3 and Fig. 9 is within the range of data presented in Fig. 3 (van der Wateren and
Dunai, 2001; Bierman et al., 2014; Kounov et al., 2015). There is no systematic spatial variation in surface lowering rates of
the pediments that can be correlated to pediment size, or geology. The Prince Albert area alluviated pediment is the most
isolated from the CFB, with no duricrust present (Fig. 4a), which can explain why the surface lowering rates are the highest in
this location (0.954 m My$^{-1}$ compared to a maximum of 0.587 m My$^{-1}$ for the other pediments). Further, the pediment surfaces
only remain fossilised as long as the duricrust remains. When the duricrust is removed denudation rates likely increase slightly
as shown by the Prince Albert area alluviated pediment, but will still remain low compared to other landforms (Fig. 3, Table
3). Therefore, the duricrusts represent an intrinsic geomorphic threshold. The $^{10}$Be-derived exposure ages of the pediments are
*minimum* estimates, and they reveal that the pediments are older than the Pleistocene, however, to further constrain this,
geomorphic and stratigraphic information needs to be integrated.

The volume of material removed by river incision into the pediment surfaces equates to a lithological thickness of 42 to 141
m (Table 4). Assuming an average maximum denudation rate of the surrounding CFB area (10.16 m My$^{-1}$ from Scharf et al.,
2013 and Kounov et al., 2015), we can estimate that the dissection started as early as ~2 to 14 Ma ago. Cosmogenic and
thermochronological (apatite fission track and (U-Th)/He) studies have reported low denudation rates across the Cenozoic,
and Scharf et al. (2013) stated that the close agreement between the CRN-based denudation and AFTA/(U-Th)/He exhumation
rates is indicative of relative tectonic stability over the last $10^6$ to $10^8$ years.

As the dissection would have occurred after the formation of the alluviated pediments, they need to be older than the start of
the incision phase (2- 14 My). Based on the observed denudation of the sub-catchments within the CFB that back the pediments
and the mean maximum denudation rates from Scharf et al. 2013 and Kounov et al. 2015 (Figs. 3 and 8, Table 5), we obtain
indicative ages of 9 - 14 My for the Laingsburg area pediment, 3 - 4 My for Floriskraal, 2 - 10 My for Leeuwgat and 2 – 28
My for Prince Albert. The CFB subcatchment denudation ages represent the ages of the dissecting rivers reaching the CFB
after dissecting the pediment surfaces. These indicative ages must be taken with caution as maximum published rates have
been used, and denudation rates vary over time, with a phase of increased erosion likely forming the incised channels.
Nonetheless, the indicative ages are useful to put the *minimum* exposure ages from cosmogenic dating in context. Furthermore,
as shown by the pediments causing the deflection of surrounding rivers (Fig. 13), denudation of the pediment material is
complicated further as the resistance of the pediment is higher than the surrounding bedrock in some locations.




Using a combination of the data above, including data on the dissection of the pediment and backing subcatchments eroded
into the resistant Cape Fold Belt Catchments, the Laingsburg area pediment could have an age of 23 Ma; Floriskraal 8 Ma;
Leeuwgat 10 Ma; and Prince Albert 17 Ma. These age estimates correspond to the timing of cessation of pediment formation
and start of dissection, and are based on the assumption that geomorphic process rates were steady over long timescales. As
denudation rates vary spatially and temporally, constant rates of erosion are unlikely as increased phases of activity are often
related to incision of the pediments. From geomorphic evidence, it is clear that the indicative ages are an order of magnitude
higher than the *minimum* exposure ages obtained from in-situ produced cosmogenic nuclide concentrations. If the cosmogenic
*minimum* exposure ages are used, with the volume eroded recorded using the DEM, erosion rates range from 28 to 503 m Ma$^{-1}$
which further indicates the minimum exposure ages should be taken with caution as these extremely high erosion rates have
not been recorded using published studies (Fig. 3). Previous works have classified pediment surfaces within height brackets
(e.g., King, 1953). However, in this study there is no correlation between pediment elevation and their geomorphic ages.

Duricrusts are found in many of the studied alluviated pediments (Summerfield, 1983; Marker et al., 2002), and this is well-
developed in the Laingsburg area pediment (Fig. 5). The alluviated pediments no longer have the overlying weathering material
preserved, and have been lowered to the iron pan layer. The depth profile suggests that deflation has occurred after the
development of the weathering mantle (Fig. 9), which has exposed the iron pan (laterites). The iron pan could have formed by
leaching from surrounding lithologies and clasts, by lateral movement due to groundwater change (Widdowson, 2007), or by
deep weathering of the bedrock. Deep weathering with the formation of iron pans occurs on low relief surfaces that have been
stable for at least a million years (Al-Subbary et al., 1998). Since the Cenozoic, South Africa has been relatively tectonically
quiescent (e.g., Bierman et al., 2014). In addition, a favourable climate of high annual rainfall, high humidity and high mean
annual temperature is required to form laterites (Widdowson, 2007). Further, higher concentrations of carbon dioxide are also
associated with the formation of laterites (and iron pans). Greenhouse episodes have occurred in the late Cretaceous and late
Palaeocene to early Eocene, leading to world-wide extensive weathering (Bardossy, 1981; Valeton, 1983).
Laterite development in southern South Africa is still poorly constrained. It has been argued to be late Pliocene in age (Marker
and Holmes, 1999) and have continued into the late Pleistocene (Marker and Holmes, 2005), being a component of the
Quaternary development of the Southern Cape (Marker et al., 2002). However, the Mediterranean climate (e.g., more humid)
of the coastal areas does not extend inland to the study location, which is expected for laterite development (Brown et al.,
1994; Braucher et al 1998a, b). Given the past climate and tectonic events, the iron pans probably formed during the late
Cretaceous greenhouse episode, which is compounded by the constrained dissection rates of the pediment surfaces (e.g.,
Dauteuil et al., 2015). The formation of duricrusts and iron pans would have occurred coevally with pediment formation, and
would have extended post-pediment formation (Helgren and Butzer, 1977; Widdowson, 2007). The presence of iron pans



indicates a period of geomorphic stability within the development of the landforms of at least 1 Ma, and probably much longer
and could have occurred during the denudation of the pediments.

**5.4 Sequence of events**

Pediment formation requires mountain range retreat, which causes the underlying lithological strata to be truncated (Figure
14). The *minimum* exposure ages calculated by cosmogenic nuclide dating using the boulder surface samples show remarkably
low denudation rates of the pediments during the last 3.8 Myr, which is related both to lithology (duricrust cappings, resistant
quartzite boulders; e.g., Scharf et al., 2013) and structure of the CFB deflecting incising rivers. The complex concentration
depth profile indicates that a recent phase of deflation has occurred, as there exists a discrepancy between the CRN
concentration of the residual boulders at the surface, and the boulders that are embedded in a sandy matrix at 30 cm depth. It
is important to integrate geomorphologic and stratigraphic knowledge when reporting cosmogenic nuclide results, especially
in an ancient setting with low denudation rates where the nuclide concentrations may reach secular equilibrium to further
extend the landscape development history.
During the Cretaceous the Cape Fold Belt was exhumed (Fig. 14; Tinker et al. 2008a, Tankard et al. 2009). During this time,
the folded strata was eroded and planed by hillslope processes (e.g., Rich, 1935; Bourne and Twidale, 1998), depositing
colluvial material and then forming soils (Fig. 14) on the alluviated pediments. This was aided by the humid climate and
greenhouse conditions of the Cretaceous causing deep weathering (Bardossy, 1981; Valeton, 1983). Tectonic stability allowed
the formation of iron pans and duricrusts, which are now exposed at the surface of the alluviated pediments due to surface
deflation and the removal of overbank material, as shown by the depth profile (Fig. 14). The initial planation and colluvial
build-up had to have occurred pre-Miocene as shown by the dissection data (Tables 4, 5). However, we posit the surfaces could
have formed much earlier due to the very slow processes associated with pediment formation (e.g., Lustig, 1969; Dohrenwend
and Parsons, 2009). By the mid-Miocene, dissection of the pediments and backing Cape Fold Belt occurred with the
development of small streams and subcatchments draining the pediments, with a shift towards a more fluvial dominated regime.
This latter stage of landscape development has decoupled the pediments from the CFB sediment source, and essentially
fossilised the landform (Table 3), with very low surface lowering (~0.6 m/My) and a more recent phase of aeolian deflation.

**5.5 Implications for landscape development**

The evolution of the pediment surfaces studied in South Africa indicate that the relative importance of hillslope and fluvial
processes (including valley development) varies over time. Therefore, the model proposed here does not fit into the previously
published model types (Fig. 1) that argued pediment evolution is dominated by a single process (e.g., 'Model 1' Figure 1;
Gilbert, 1877; Paige, 1912; Howard 1942 and 'Model 2' Fig. 1; Lawson, 1915; Rich; 1935; Kesel, 1977; Bourne and Twidale,
1998; Dauteuil et al., 2015), dominance varies due to lithology (e.g., 'Model 3' Figure 1: Lustig, 1969; Parsons and Abrahams,
1984) or is assisted by valley / basin development (e.g., 'Model 4' Fig. 1; Lustig, 1969; Parsons and Abrahams, 1984). The





change from hillslope to fluvial processes is likely a response to tectonic or climatic perturbations (Fig. 14). The initial
formation of the pediments was most likely aided by large-scale erosion during the Cretaceous (e.g., Tinker et al., 2008a,b;
Wildman et al., 2015, 2016; Richardson et al., 2017) and tropical conditions (Partridge and Maud, 2000).
The indicative geomorphic ages reported here, related to the second phase of development and the dissection of the pediments
by small tributaries, roughly correlate to the proposed uplift in the Cenozoic (Green et al., 2016) of 30 Ma (Burke, 1996), 18
Ma (Partridge and Maud, 1987) and 2.5 Ma (Partridge and Maud, 1987), and could indicate that the pediments were dissected
due to different pulses of uplift. Nonetheless, this time period also corresponds to variation in climate, including periods of
humidity reported to have ended at 30 Ma (Burke, 1996) or 18 Ma (Burke, 1996). It is not possible to distinguish the main
driver of dissection, and tectonic signatures are not identified within the Gouritz catchment morphometry (Richardson et al.,

537  2016).

The grading of the pediments indicates the main trunk rivers were active before the pediments, at least by the Miocene and
probably established within the Cretaceous, when large scale exhumation occurred within South Africa (e.g., Tinker et al.
2008a, Richardson et al., 2017). The individual grading of the pediment surfaces indicates the pediments are relatively local
features that react to surrounding tectonic, geological, and geomorphological settings, and are not singular surfaces (King,
1953). The surface lowering rates of the pediments indicate a period of low geomorphic activity as documented by other
researchers (Fig. 3, and references therein). There has been a drastic reduction in denudation rates since the Cretaceous as
shown by apatite fission track and cosmogenic nuclide studies (Fig. 3 and references therein). The data reported in this study
are some of the lowest in the world (Portenga and Bierman, 2011). Surface lowering is not consistent across landforms within
southern South Africa. Rivers are dissecting at a faster rate (Scharf et al., 2013; Kounov et al., 2015) than the pediment surfaces
(this study, van der Wateran and Dunai, 2001; Bierman et al., 2014; Kounov et al., 2015), which indicates that relief is
developing at a slow rate, as also reported by Bierman et al. (2014) from the Eastern Cape. The offshore depositional record
(Tinker et al. 2008a) mirrors the reduction in denudation rates with peaks in the Cenozoic most likely related to the rejuvenation
of the landscape, which dissected the pediments in this study (e.g., Hirsch et al., 2010; Dalton et al., 2015; Sonibare et al.,
2015). These increases in offshore sediment flux are minor in comparison to rates in the Cretaceous.
**6. Conclusion**
Large-scale erosional surfaces characterise the ancient landscape of southern South Africa. Cosmogenic nuclide dating using
$^{10}$Be of four pediment surfaces in the Western Cape, and a depth profile indicate low surface lowering rates (0.315 to 0.954 m
My$^{-1}$) and *minimum* exposure ages from the Early Pleistocene. Given that the isotope concentrations are close to isotopic steady
state, the $^{10}$Be-derived exposure ages are *minimum* estimates. Cosmogenic radionuclide depth profiling revealed that the post-
depositional history of the alluviated pediments is likely to be complex, with a long period of slow denudation that is followed
by a phase of aeolian deflation. Further work, beyond the scope of this study is needed to understand if this is a widespread



and characteristic feature of alluviated pediment surfaces in (semi-)arid climatic conditions. The pediments studied must be at
least Miocene in age, and probably much older (i.e. Cretaceous) based on the volumes of post-pediment dissection, published
erosion rates, the presence of duricrusts and the current understanding of tectonic and climatic variation in the region. The
duricrusts represent an internal geomorphic threshold which limits the rate of denudation. The dissection of the pediments has
been largely controlled by the structure of the Cape Fold Belt, with the initial geomorphic pulse of incision most likely related
to tectonic uplift or climate change. The pediments grade to individual base levels (trunk rivers), and although locally extensive,
they are not a regional feature representing one single surface. The presence of the pediments deflected dissecting rivers in
some locations and controlled landscape evolution of the surrounding rivers.
The pediments in southern South Africa are lowering at very low rates and are now decoupled from the surrounding rivers.
Therefore, they are a fossilised landform that represents a relatively stable store of sediment in which surface lowering occurs
by aeolian erosion causing deflation. The persistence of the pediments is due to the resistant duricrust capping and quarzitic
boulders, and the structural control of the Cape Fold Belt and pediments, deflecting dissecting rivers. We contend that
cosmogenic nuclide results must not be viewed in isolation and should be assessed together with surrounding geomorphologic
and stratigraphic conditions.

**Author Contributions**

Janet C. Richardson, David Hogdson and Andreas Lang collected the data. Processing and analysis of the data was completed
by Janet C. Richardson and Veerle Vanacker. Marcus Christl measured the $^{10}$Be/$^9$Be using an accelerator mass spectrometer
on the 500 kV Tandy facility at ETH Zürich. Veerle Vanacker provided further support processing the data with regards to the
depth profile, creating Figure 9 and writing the methodology for cosmogenic nuclides. Janet C. Richardson led the writing and
drafting of figures, with contributions on the text and figures by Veerle Vanacker, David Hodgson and Andreas Lang.

**Acknowledgements**

The British Geomorphology Society (BSG) and British Sedimentology Research Group (BSRG) are thanked for providing
postgraduate grants to J. Richardson for completing this research. Jérôme Schoonejans and Marco Bravin are thanked for their
help during laboratory work undertaken in Université catholique de Louvain, Belgium. David Lee is thanked for his help in
improving Fig. 1. The landowners in South Africa are thanked for their permission to enter their land and take samples. The
Council of Geoscience are thanked for providing Geology GIS tiles, under the Academic/Research license. Alexandre Kounov
and an anonymous reviewer are thanked for their reviews of a previous version of this paper.



**Competing interests**

Andreas Lang is a member of the editorial board for Earth Surface Dynamics.

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
