# Peer review of "Constraining the timing and processes of pediment formation and"

_EGUsphere, 2024_

## Referee Comment (RC2)

[referee-annotated manuscript omitted]

---

## Author Comment (AC1)

'Constraining the timing and processes of pediment formation and dissection: implications for long-term evolution in the Western Cape, South Africa'

We would like to thank both reviewers for their positive reviews and very helpful suggestions to improve the manuscript. We have replied individually below.

**Reviewer 1: R. Braucher**

The paper of Richardson et al. clearly presents attempts to constrain the timing and processed involved in pediment formation in South Africa.

The authors present different models of pediment formation based on literature then they show their sampling sites, and their chronology based on $^{10}$Be measurements and finally discussed their results.

I will have only some questions on the cosmogenic part.

- Top sample of depth profile is not at a different altitude than the other samples below (779 and 776 m); Is this correct? (table 1) Is the density of the top profile sample also set at 1.6 ?

Thank you for spotting this; the text has been updated accordingly for altitude and density in table 1.

- I was able to redo all calculations and agree with minimum ages and maximum denudation rates determined by the authors (see joined excel file).

Thank you for supplying an excel file and for independently cross-checking the denudation rate calculations.

- However, I disagree with their exposure ages determined with 0.3 m/My denudation rate. Based on $^{10}$Be only I am sure that ages up to 4.6My can be determined (table 3 last column). Using the integration time formula of Lal (1991) involving neutrons only I have an upper age of 1.04 My ( see excel All samples sheet)

We are happy to look into these issues. The ages reported in Table 3 are minimum exposure ages for non-eroding and slowly eroding profiles, with minimum ages of 0.315 My for non-eroding surfaces.

- Regarding the depth profile I suggest the authors to do some more investigations. From this depth profile, sample SA-LB-DP0 is an outlier compared to the other samples. Considering all samples I determined a "slope decrease" of 112 g/cm2 in disagreement with neutrons attenuation length ( ~160g/cm2); Removing the upper samples the "slope" is 155.6 g/cm2 in agreement with low denudation and neutrons attenuation.
- From this depth profile, minimum ages can be determined for each point as well as max denudation rates. This can show that the profile is almost at steady state state.
- Then it will be nice if the authors could model their depth profile to determine denudation rate and exposure time. I reach a minimum age of 360ka with no denudation.
- Now because the deepest sample exhibits the higher maximum denudation rate (5.47 m/My based-on my calculation), this sample can be used to better estimate the exposure age of the entire profile. To do so, I have considered a denudation of 3.1 m/My for all samples (based on max denudation rate of sample SA-LB_DP30 ) and infinite time for all samples except the deepest one. An exposure time of 867ka was achieved for the deepest. This age can better reflect the true age of the profile.

We thank the reviewer for these comments. We agree that the upper sample site from the depth profile represents an outlier, with the concentration of the sample reflecting a sample higher up on the depth profile. Because of this, we interpret that deflation has removed fine material, and that the sampled boulder is part of a deflation surface. We also agree that the 'slope' of the concentration-depth profile

(after removing the sample from the deflation surface) is in very well agreement with the expected path attenuation length in these sediments. We will further elaborate this point in the revised manuscript.

Also, we will improve the concentration-depth model, and provide maximum denudation rates, and minimum exposure ages for the profile.

- The authors suggest a denudation change in the past; This has already be evidenced with cosmo but with two nuclides ( [10]Be and 16Al) ( Jolivet et al. 2021 https://dx.doi.org/10.1016/j.geomorph.2021.107747 ; Godard et al 2021. ⟨1002/esp.5190⟩. I think that using one nuclide this is more difficult. I tried to model this with a denudation change from 0.3 m/My to 3.1m/Ma; I can reach a n age of 3My but the 30cm deep sample from the depth profile is not well modeled.  Why did the authors not measure $^{26}$Al? This can be a real nice input and help to see a denudation rate change. I will not ask for that measurements for this paper but if the authors still have some remaining fractions, I encourage them to test (eventually they can contact me if help is needed).

Thank you for the suggestions and comment. We have read the suggested citations and we will integrate these into the text in the appropriate places. At the moment that we processed the samples, we did not consider it relevant to measure in-situ 26Al concentrations, as complex burial was not expected. We will consider this for future work and thank you for your offer of help.

- I suggest accepting this paper with minor revisions

Thank you.

**Reviewer 2: A. Kounov**

Dear Editor,

I reviewed this manuscript a few years ago. Upon reading the recent version, I have noticed some positive improvements. Therefore, I can only reiterate what I concluded about the manuscript previously.

The manuscript is well written and scientifically interesting. I think that the presented in this study data generally well support the suggested conclusions and the presented sequence of events during the evolution of the studied geomorphological features (Fig. 14). This study brings important advances in the better understanding of the Cretaceous and Cenozoic landscape evolution of South Africa. It also takes a significant step forward in challenging the paradigm of the existence of old, singular, large-scale erosional surfaces in southern Africa.

I have annotated a PDF copy of the manuscript with some minor comments.

Finally, I would recommend the publication of this manuscript after only minor corrections.

Kind regards.

Alexandre Kounov

We would like to thank you for reviewing the updated version of manuscript, and for your constructive comments for both reviews that have led to improvements.

Annotated pdf comments:

We have updated the text, thank you for spotting the mistake [line 56-58]:

Tracked changes:  Model type 1 acknowledges the occurrence of channelised processes and model type 2 acknowledges the occurrence of diffusive erosion processes, but each model argues these are subsidiary formation processes

Section 2.1 Geological setting comments – We have updated the text to include the information on: ages of supergroups, metamorphism history, intrusion information and date of tectonic shortening, location/timing of denudation, and why some studies exclude Cenozoic uplift [lines 98-118]:

Tracked changes:  In the  Western Cape, Southern Africa, the geology is dominated by strata of the Cape (Early Ordovician to Early Carboniferous and Karoo Supergroups (Late Carboniferous to Early Jurassic) (Johnson et al. 1995, Frimmel et al. 2001) (Fig. 2), which are composed of various sandstone, siltstone and mudstone successions. Both supergroups have been subject to low-grade burial metamorphism (Frimmel et al., 2001), with localised contact metamorphism during Jurassic dolerite intrusion (Johnson et al.1995), and an estimated 6-7 km of exhumation during the Early Cretaceous (Tinker et al., 2008; Wildman et al., 2015).. Tectonic shortening during the latest Paleozoic-to-early Mesozoic of the Cape and Karoo Supergroups  (Tankard et al. 2008; Hansma et al. 2016) have resulted in  E-W trending, northward verging, and eastward plunging folds that decrease in amplitude northward and shorten northwards, and form the backbone of the exhumed Cape Fold Belt (CFB) (Paton, 2006; Tinker et al., 2008b; Scharf et al., 2013; Spikings et al., 2015). During the Mesozoic, the rifting of Gondwana initiated large-scale denudation across southern Africa. Using apatite fission track analyses of outcrop and borehole samples, Tinker et al.(2008a) concluded that the southern Cape escarpment and coastal plain underwent 3.3 to 4.5 km of denudation since the mid-late Cretaceous and potentially 1.5 to 4 km within the early Cretaceous, using a thermal gradient of ~20°C/km. Wildman et al. (2015) processed 75 apatite fission track and 8 zircon fission track data from outcrop and boreholes across the southwestern cape of South Africa (from coast to the escarpment). Using a thermal history model and a thermal gradient of 22°C/km, they obtained an average of 4.5 km of denudation in the Mesozoic since the Late Jurassic-Early Cretaceous. However, the estimates range between 2.2 and 8.8 km of denudation using the upper and lower ranges of the geothermal gradient and possible thermal histories bounded by 95% significance intervals, which provides uncertainty on the inferred model. Richardson et al. (2017) used reconstructed geological cross sections, tied to apatite fission track data, and drainage reconstruction to model up to 4-11 km of denudation across the Western Cape, with significant exhumation in the Early Cretaceous and lower amounts in the Late Cretaceous.

Figure 2 – The figure caption has been updated – line 123:

Tracked changes: **Figure 2: Stratigraphic chart showing the major lithostratigraphic units of the Western Cape, South Africa.**

Figure 3 – We have updated the figure to help support the interpretation. We have also prepared some supplementary information so that the data can be viewed in table format.

[Figure]

Section 4.2 – We have updated the text to reflect the comments regarding why we chose an erosion rate of 0.3m/myr and minimum ages:

Tracked changes: Assuming low erosion rates of 0.3 m My$^{-1}$ (following Bierman et al. 2014) the pediment minimum exposure ages increase substantially for the older surfaces, with minimum ages ranging from 0.678 to 4.462 My (Table 3).

 [Line 303-305]

Section 5.2 comment around deflation. The 0.6m/Myr relates to the whole profile, whereas the top depth profile sample has increased concentrations than expected. We interpret that deflation has removed material and there are missing data points, related to different geomorphic processes functioning over geological time. We will further elaborate the text on the concentration-depth profile following Brauchers' comments.

Section 5.3 – We have updated the text following the reviewers comments around the processes and occurrence of pediments and fluvial networks to improve the clarity of the text and to acknowledge the assumptions in our geomorphic interpretation.

Tracked changes: Based on the observed denudation of the sub-catchments within the CFB that back the pediments and the mean maximum denudation rates from Scharf et al. 2013 and Kounov et al. 2015 (Figs. 3 and 8, Table 5), we obtain indicative minimum ages of 9 - 14 My for the Laingsburg area pediment, 3 - 4 My for Floriskraal, 2 - 10 My for Leeuwgat and 2 – 28 My for Prince Albert. [Line 464 – 467]

Tracked changes: These age estimates correspond to the timing of cessation of pediment formation and start of dissection, and are based on the assumption that geomorphic process rates were steady over long timescales. [line 477-478]

---

## Author Response (AR1)

Reply to reviews

We would like to thank both reviewers for their positive reviews and very helpful suggestions to improve the manuscript. We have replied individually below.

**Reviewer 1: R. Braucher**

The paper of Richardson et al. clearly presents attempts to constrain the timing and processed involved in pediment formation in South Africa.

The authors present different models of pediment formation based on literature then they show their sampling sites, and their chronology based on [10]Be measurements and finally discussed their results.

I will have only some questions on the cosmogenic part.

- Top sample of depth profile is not at a different altitude than the other samples below (779 and 776 m); Is this correct? (table 1) Is the density of the top profile sample also set at 1.6 ?

Thank you for spotting this. This typo has been corrected (776 instead of 779 m a.s.l.) in Table 1. As you noticed correctly, we decided to take a density of 1.6 g/cm³ for the top sample of the depth profile. We consider that this sample was exposed at the surface after the deflation of the finer material of the matrix in which the boulder was embedded. For the top samples, the density of the overburden does not affect the age determination.

- I was able to redo all calculations and agree with minimum ages and maximum denudation rates determined by the authors (see joined excel file).

Thank you for supplying an excel file and for independently cross-checking the denudation rate calculations.

- However, I disagree with their exposure ages determined with 0.3 m/My denudation rate. Based on [10]Be only I am sure that ages up to 4.6My can be determined (table 3 last column). Using the integration time formula of Lal (1991) involving neutrons only I have an upper age of 1.04 My ( see excel All samples sheet)
-

We would like to clarify that the in-situ [10]Be based exposure ages that we report in Table 3 are 'minimum exposure ages'. We report them for a scenario where we assume (1) zero erosion and no burial (Table 3, 5th column) and (2) steady erosion of 0.3 m/My (Table 3, 6th column). The exposure ages that we obtained with the CosmoCalc 3.0 version are very similar as the ones provided by the reviewer, with minimum exposure ages between 0.569 and 1.377 Ma for the no-erosion scenario and between 0.678 and 4.462 Ma for the 0.3 m/My erosion scenario.

- Regarding the depth profile I suggest the authors to do some more investigations. From this depth profile, sample SA-LB-DP0 is an outlier compared to the other samples. Considering all samples I determined a "slope decrease" of 112 g/cm2 in disagreement with neutrons attenuation length ( ~160g/cm2); Removing the upper samples the "slope" is 155.6 g/cm2 in agreement with low denudation and neutrons attenuation.

We appreciate this remark and agree with the reviewer that the exponent of the exponential model fit allows us to derive the path attenuation during in-situ 10Be production. Based on the lowermost four data points, we obtain an exponential model fit with an exponent of -0.01, corresponding to a path attenuation of 160 g/cm² for an overburden with a density of 1.6 g/cm³. These values are perfectly in line with the path attenuation length for neutron spallation as pointed out by the reviewer. The in-situ 10Be concentration of the boulder at the surface is more than double the theoretically expected concentration from the exponential model fit, and can be attributed to surface deflation. We estimate that about 110 cm of fine material was removed from the top of the pediment, forming deflation armouring. We further developed this argumentation in the section 4.2

- From this depth profile, minimum ages can be determined for each point as well as max denudation rates. This can show that the profile is almost at steady state state.
- Then it will be nice if the authors could model their depth profile to determine denudation rate and exposure time. I reach a minimum age of 360ka with no denudation.
- Now because the deepest sample exhibits the higher maximum denudation rate (5.47 m/My based-on my calculation), this sample can be used to better estimate the exposure age of the entire profile. To do so, I have considered a denudation of 3.1 m/My for all samples (based on max denudation rate of sample SA-LB_DP30 ) and infinite time for all samples except the deepest one. An exposure time of 867ka was achieved for the deepest. This age can better reflect the true age of the profile.

We thank the reviewer for these suggestions, and we used forward modelling approaches (as developed for Vandermaelen et al. (2022)) to explore the erosion – exposure age scenarios that can best explain the observed $^{10}$Be depth concentrations in the Laingsburg pediment. The goodness-of-fit of the model predictions was evaluated based on the Nash-Sutcliffe model efficiency (NSE) and minimising chi-square, by comparing modelled with measured $^{10}$Be concentrations for the 5 samples.

We ran simulations for a wide spectrum of erosion (0 to 1.5 m/My) and exposure age (0 to 20 Ma), and for conditions with/without inheritance, and with/without deflation armoring. The outcomes of the model predictions are now summarised in Fig. 9b and Fig.11, and discussed in the text in section 4.2 and 5.2. They show that the measured $^{10}$Be depth concentration profile cannot be modelled well (NSE < 0.6) when deflation is not accounted for. Optimal model solutions were found for long-term erosion rates between 0.3 and 0.6 m/My and exposure age exceeding 2 Ma. The model predictions also reveal that the samples approach isotopic steady state, with $^{10}$Be concentrations becoming time-invariant for E=0.6 m/My.

This has been further elaborated in the discussion in section 5.2, and the abstract and conclusion have been modified accordingly to include the results of the forward modelling exercise.

- The authors suggest a denudation change in the past; This has already be evidenced with cosmo but with two nuclides ( $^{10}$Be and 16Al) ( Jolivet et al. 2021 https://dx.doi.org/10.1016/j.geomorph.2021.107747 ; Godard et al 2021. ⟨1002/esp.5190⟩. I think that using one nuclide this is more difficult. I tried to model this with a denudation change from 0.3 m/My to 3.1m/Ma; I can reach a n age of 3My but the 30cm deep sample from the depth profile is not well modeled.  Why did the authors not measure $^{26}$Al? This can be a real nice input and help to see a denudation rate change. I will not ask for that measurements for this paper but if the authors still have some remaining fractions, I encourage them to test (eventually they can contact me if help is needed).

We appreciate this suggestion and will take it along for future work. At the time of sampling and sample processing, we did not expect to have a complex erosion-exposure age history, so we processed samples for $^{10}$Be only. We see the added value of processing them for $^{26}$Al now more clearly, and will consider this in future work.

- I suggest accepting this paper with minor revisions

We appreciate the very helpful and constructive comments which stimulated us to realise further analyses and simulations of the data.

**Reviewer 2: A. Kounov**

Dear Editor,

I reviewed this manuscript a few years ago. Upon reading the recent version, I have noticed some positive improvements. Therefore, I can only reiterate what I concluded about the manuscript previously.

The manuscript is well written and scientifically interesting. I think that the presented in this study data generally well support the suggested conclusions and the presented sequence of events during the evolution of the studied geomorphological features (Fig. 14). This study brings important advances in the better understanding of the Cretaceous and Cenozoic landscape evolution of South Africa. It also takes a significant step forward in challenging the paradigm of the existence of old, singular, large-scale erosional surfaces in southern Africa.

I have annotated a PDF copy of the manuscript with some minor comments.

Finally, I would recommend the publication of this manuscript after only minor corrections.

Kind regards.

Alexandre Kounov

We would like to thank you for reviewing the updated version of manuscript, and for your constructive comments for both reviews that have led to improvements.

Annotated pdf comments:

Typos / clarifications and smaller changes:

We made the necessary modifications in the text, and clarified the description of the escarpment retreat models [line 203-205], and provided more details on the overall geological setting [Lines 245 – 205]. We updated the caption of Figure 2 and have updated Figure 3 to make it more readable.

We have updated the text around uplift to reflect the difference between mantle plumes and epierogenic uplift and have included the dispute around Cenozoic uplift [Lines 273-281].

[Line 303-305] – Choice of 0.3 m/My for erosion rate.

For the calculations of the minimum exposure ages with CosmoCalc, we provided results for a non-eroding surface and a 0.3 m/My eroding surface. The choice of 0.3 m/My is based on earlier work of Bierman et al. (2014), and our model simulations show that models predicting 0.3 – 0.6 m/My of erosion have the highest model performance. We have revised section 4.2, and included new results from model predictions that include a range of erosion-exposure age scenarios. The outcomes of the model simulations are shown in Fig 11, and their implications are discussed in section 5.2.

Section 5.2 comments on deflation.

We have clarified section 5.2 and 5.3 and strengthened this section by incorporating the results of model simulations. The erosion rates refer to the long-term surface erosion over the time of exposure. Given the > 2 million-year exposure of the pediment, the erosion rate is very slow and can have taken place during soil formation. The four lowermost points of the depth profile show a steady-state $^{10}$Be depth concentration profile, with a path attenuation congruent with the theoretically expected one for steady-state profiles. The development of the armor layer postdates this phase, and is recent compared to the

buildup of the steady-state profiles. A dual isotope approach (with $^{10}$Be and $^{26}$Al measured on the same samples) could be informative to resolve the period of recent deflation.

Section 5.3 comments on processes.

We have updated the text following the reviewers comments around the processes and occurrence of pediments and fluvial networks to improve the clarity of the text [line 683] and to acknowledge the assumptions in our geomorphic interpretation [lines 694 – 695].

---

## Author Response (AR2)

AE comments (black), author replies (green):

The authors have satisfactorily addressed reviews and clarified important assumptions related to their calculation of denudation rate. Some very technical corrections may be in order.

First, please ensure that the values that appear in Tables 4 and 5 (and throughout the text) are shown with the appropriate number of significant figures.

We have gone through the manuscript to ensure consistency with significant figures and have reflected the updates in the text (tracked changes in MS below).

Second, please check that Line 18 in the abstract reads correctly.

We have updated this and added the missing word.

Finally, Texp is defined in a figure caption late in the paper as exposure age for the first time. Please check that the introduction of this new term is necessary.

We have introduced this term earlier in the methods section (Line 290)

[revised manuscript text omitted]

**Table 3: Cosmogenic nuclide data for surface samples from pediments. The reported [10]Be concentrations are corrected**
**for procedural blanks, using a value of (7.54 ± 9.67) × 10^{−15}, and the 1σ uncertainty estimates contain**
**analytical errors from AMS measurement and blank error propagation. Maximum denudation rates and minimum**
**durations of surface exposure were calculated using the CosmoCalc add-in for Excel (Vermeesch, 2007). For the surface**
**exposure ages (Texp), we assumed (1) no erosion or burial since exposure, and (2) a maximum steady erosion rate of**
**0.3 m My^{-1}.**

[revised manuscript text omitted]